# Factors Associated with Caregiver Burden in Caregivers of Older Patients with Dementia in Indonesia

**DOI:** 10.3390/ijerph191912437

**Published:** 2022-09-29

**Authors:** Yossie Susanti Eka Putri, I Gusti Ngurah Edi Putra, Annida Falahaini, Ice Yulia Wardani

**Affiliations:** 1Department of Mental Health and Psychiatric Nursing, Faculty of Nursing, Universitas Indonesia, Depok 16424, Indonesia; 2Institute of Population Health, University of Liverpool, Liverpool L69 7ZA, UK; 3Suradadi Hospital, Tegal 52182, Indonesia

**Keywords:** burden, caregiver, dementia, social support, BPSD, Indonesia

## Abstract

This cross-sectional study aimed to identify caregiver burden and its determinants in the informal caregivers of older patients with dementia (PWDs) aged ≥ 60 years in Java, Indonesia. Data were collected from 207 caregivers of older PWDs using self-administered questionnaires. The dependent variable was caregiver subjective burden, assessed using the Zarit Burden Interview (ZBI). The independent variables included the socio-demographic characteristics of PWDs and caregivers, the caregiver’s perceived social support, and the behavioural and psychological symptoms of dementia (BPSD). Linear regression with a stepwise elimination method was used to identify the factors associated with caregiver burden. This study found that four factors were associated with the caregiver burden, such as the gender of PWDs, the educational level of caregivers, social support, and BPSD (R-squared = 27.78%). Higher burden was reported among the caregivers of female PWDs (β = 5.58; 95%CI = 2.16; 8.99) and PWDs with higher scores of BPSD (β = 0.34; 95%CI = 0.25; 0.43). Meanwhile, the caregivers with higher perceived social support (β = −0.26; 95%CI = −0.42; −0.10) and who completed high school education and above (β = −6.41; 95%CI = −10.07; −2.74) tended to have lower scores of subjective burden. These findings suggest that BPSD management and maintaining the resources of support may provide an opportunity to minimise caregiver burden and improve the quality of life of caregivers and PWDs.

## 1. Introduction

As the fourth most populous country worldwide, Indonesia has experienced an increase in life expectancy at birth in the last decade, from 67.89 and 71.83 years in 2010 to 69.67 and 73.55 years for males and females in 2021, respectively [1]. In 2020, the percentage of older adults (60 years and older) was around 11% of the total population, and it is estimated to nearly reach 20% in 2045 [2]. It is conceivable that the prevalence of ageing-associated diseases, such as dementia, is expected to be an impact of these gradual changes in the Indonesian population structure. According to the report by Alzheimer’s Disease International [3], Indonesia ranked ninth globally for the number of people with dementia, accounting for 1.2 million people affected in 2015.

Dementia is a general term to describe the symptoms that occur when the brain is affected by certain diseases or conditions and is one of the most common degenerative diseases in older adults [4]. The progressive development of dementia and its accompanying pathological changes impact the cognitive, memory, and behaviour of older adults. This condition significantly affects the ability of the elderly to carry out daily activities, so they need a companion in their daily life [5]. In such circumstances, a caregiver is at the forefront in caring for older patients with dementia (PWDs), especially in Indonesia, where healthcare services for dementia care are limited. As a result, the healthcare system relies heavily on informal caregivers to support the care for PWDs, which falls upon family members such as spouses, children, and relatives. However, concern regarding caregivers is still limited; thus, challenging issues among caregivers are undetected.

The caregiving role has been traditionally viewed as a woman’s responsibility in Indonesia, similar to many countries, especially when the care recipients are parents. In addition, the findings from an ethnography study in East Java and West Sumatra demonstrated the preference of a daughter to fulfil the caring responsibility of older adults [6]. Caring for the infirm members of the household is considered a cultural duty for the whole family members in the Indonesian society, which is also in line with religious values. Being a caregiver of their loved one might be interpreted as gratitude among caregivers. This also reaffirms the emotional bond between them, and it is also perceived as a natural consequence of kinship ties and moral values [7]. It is a way to show devotion and affection to the care recipients. Strong kinship ties can diminish the level of caregiver stress. However, on the other hand, caring for older PWDs also poses challenges for family members who care for them because dementia is progressive; the dependency level of dementia patients is continuously higher. This condition will trigger the burden of caring for caregivers. Burden arises because of an imbalance between the caregiver’s perceptions of demands and resources. Demands may come in the form of caregiving responsibilities, other family needs, society, and work. As such, this burden can lead to a decline in the health of caregivers and their ability to provide good care [8,9]. A study by Borges in Martins et al. [10] noted that about 87.9% of the caregivers presented complications in their health after caring for older adults, as evidenced by at least one disease. Unfortunately, PWDs and their caregivers are not considered a major public health concern in Indonesia, resulting in little attention paid to these issues.

The socio-demographic and caregiving-related factors associated with caregiver burden are widely accepted, and these factors are complex and multidimensional, affecting caregivers differently throughout the disease process. Previous studies reported that caregiver characteristics such as gender, educational attainment, employment status, relationship with the care recipient, and staying with the care recipient were predictors of caregiver burden [11,12,13,14]. In terms of care-recipient factors, previous studies reported that age, gender, and the presence of neuropsychiatric symptoms were the factors affecting caregiver burden [11,14]. However, some of these factors are not definitive in predicting caregiver burden since inconsistencies exist. For example, a study among the Croatian caregivers of PWDs found that the gender of PWDs, the duration of caregiving, and caregiver relationship predicted caregiver burden [15]; however, in Taiwanese dementia caregivers, the gender of PWDs, educational attainment, marital status, and employment status were not significant factors [12]. In the dementia caregiving literature, social support is another factor significantly contributing to caregiver burden and is perceived as a protective factor among dementia caregivers [16,17].

Given its large population of older people and its rapidly changing demographic and reform healthcare services, it is vital to examine the caregiving burden among the informal caregivers of older adults with dementia in Indonesia. The extensive research conducted on these factors across countries shows a dearth of studies in the literature on Indonesia. A lack of research on dementia caregivers in Indonesia means uncertainty exists regarding the extent of the burden reported among the caregivers of older PWDs, the characteristics of caregivers and care recipients, as well as the protective factors of caregiver burden. Understanding these issues is required to make appropriate interventions in dementia caregiving in Indonesia. This study aimed to investigate the factors associated with caregiver burden among the informal caregivers of PWDs aged ≥ 60 years in Indonesia.

## 2. Methods

### 2.1. Study Design and Sample

This was a cross-sectional study, conducted in Java, the most populous island in Indonesia, which aimed to investigate the factors associated with caregiver burden in the caregivers of older PWDs. The data for this present study were retrieved from a previous study project on the predictors of depressive symptoms among the caregivers of PWDs in Java, Indonesia [18]. In brief, the data were collected from the caregivers of PWDs who attended outpatient services in four tertiary hospitals that provide treatment for PWDs, representing four populous provinces in Java, Indonesia.

Dementia status was determined by a diagnosis from a neurologist or psychiatrist. The caregivers of PWDs were recruited using the convenience sampling technique. The inclusion criteria for the caregivers included those (1) aged 18 years or more, (2) who were the primary caregivers of PWDs for at least six months and did not receive any salary or reward for the care they provided, (3) who lived in the same house as PWDs, and (4) who were able to read and speak the Indonesian language (Bahasa). If the potential participants or caregivers were over 60 years old, the researcher or research assistant screened them for cognitive function. Ten participants who had cognitive impairment with a total score > 7 from a Six-item Cognitive Impairment Test (6CIT) were excluded from this study. In total, 207 caregivers who provided care for older PWDs aged 60 years or more were included in this present study. The participants provided written informed consent prior to participating in the study. If the participants had difficulties in completing the questionnaires, the researcher or research assistant would help them to have all the questions completed. 

### 2.2. Variables

The dependent variable in this study was caregiver burden, measured using the Zarit Burden Interview (ZBI) [19]. This measure was found to have excellent psychometric properties with a Cronbach’s alpha coefficient of 0.91 in a previous study using the same sample as that of the current study [18]. ZBI consists of 22 items that assess the burden experienced by caregivers on several aspects of their lives: health, personal life, social, financial circumstances, etc. A possible total score ranges from 0 to 88 with a higher score indicating an increase in caregiver burden. We received permission to use this instrument from the MAPI Research Trust. This instrument was translated from English into Bahasa and then back-translated by other translators. 

We modified a conceptual framework on the factors associated with caregiver burden proposed by a previous study [20]. The independent variables that were examined in this study included the socio-demographic characteristics of PWDs and caregivers, social support, and the behavioural and psychological symptoms of dementia (BPSD). The variables representing the socio-demographic characteristics of PWDs consisted of age, gender, educational level, marital status, the duration of being diagnosed with dementia, and health insurance ownership. In addition to the basic socio-demographic characteristics, the caregivers were also asked to provide information on their employment status, family monthly income, the duration of caregiving, relation to PWDs, and whether they provided care for other family members. 

The perceived social support from family, friends, and significant others was assessed using the 12-item questionnaire of the Multidimensional Scale of Perceived Social Support (MSPSS) [21]. Higher social support perceived by caregivers is indicated by a greater score with a possible range of 12 to 84 for the total score. This instrument was reported to have a Cronbach’s alpha coefficient of 0.87 [18]. We translated this scale into Bahasa and then back-translated by other translators. 

The Neuropsychiatric Inventory Questionnaire (NPI-Q), a shorter version of the Neuropsychiatric Inventory (NPI), was used to evaluate BPSD [22]. This instrument consists of 12 domains that assess the severity level of neuropsychiatric symptoms of PWDs (e.g., delusions, hallucinations, dysphoria, anxiety, etc.) and the effect of the symptoms on caregiver distress. The total scores range from 0 to 36 and 0 to 60 for severity and caregiver distress, respectively. Higher total scores from adding together both the severity and caregiver distress indicate a greater level of severity and caregiver distress. The Cronbach’s alpha coefficient for this instrument was 0.90 [18]. Similar to previous scales, we also translated this scale from English to Bahasa, and other translators translated it back.

### 2.3. Data Analysis

Descriptive statistics were used to present the socio-demographic characteristics of PWDs, caregivers, and the level of social support, BPSD, and caregiver burden. Mean and standard deviation (SD) were used to present the descriptive findings of continuous variables, whilst frequency and percentage were used for the categorical variables. Simple linear regression was used for bivariate analysis, which aimed to examine the unadjusted associations between each independent variable and the dependent variable (i.e., caregiver burden). 

Multiple linear regression was then employed to assess the adjusted associations. We used a “stepwise” elimination method to develop the multivariate model. The initial model was generated by including all the variables simultaneously in the model, irrespective of the significant level (*p*-value) from the bivariate analyses. The variable with the highest *p*-value was then excluded one at a time, followed by re-running the model after the deletion of that variable. All the variables with a *p*-value < 0.05 were retained in the model. The variables that had been deleted before were again included one a time in the model to re-examine their influence. The final multivariate model consisted of those variables with a *p*-value < 0.05. The study results are presented as regression coefficient (β) along with its 95% confidence intervals (CIs) and *p*-values.

### 2.4. Ethics Approval and Consent to Participate

The primary data collection used in this study obtained ethics approval from the Ethics Committee of Nursing Research, Faculty of Nursing, Universitas Indonesia, Indonesia (No. 138/UN2.F12.D/HKKP.02.04/2019) and the Institutional Review Board of Faculty of Nursing, Mahidol University, Thailand (No. IRB-NS 2019/488.0603). The Ethics Committee at each hospital permitted the data collection. Before participating in this study, the participants (caregivers) were provided with information about the background and aim of the study, the benefits and risks, and the compensation they would receive for their participation. In addition, the participants were also informed about their rights to withdraw from this study at any time without any consequences. Written informed consent was given by all the participating caregivers. All methods were carried out in accordance with relevant guidelines and regulations.

## 3. Results

Table 1 presents the socio-demographic characteristics of the PWDs and caregivers. While the majority of the PWDs were aged 60–74 years (72.95%), most of the caregivers were aged < 60 years (69.57%). More than half of the PWDs were males (56.04%), but the caregivers were predominantly females (74.88%). The majority of the PWDs and caregivers had at least a high school education level and were currently married at the time of the data collection. About half of the caregivers reported at the time of the data collection that they started their care less than two years since the PWDs were diagnosed and the duration of care they provided for the PWDs was less than two years, accounting for 53.62% and 55.56%, respectively. The caregivers were mostly adult children or daughters-in-law and spouses of the PWDs. We found no son-in-law providing care for the PWDs. Only a few caregivers reported that they also provided care for other family members (8.70%). Only 6.67% of the PWDs did not have health insurance.

Table 2 shows the average scores of the social support, BPSD, and caregiver burden. The level of social support perceived by the caregivers was moderate to high (mean = 65.48/84) with the highest support received from family members (mean = 23.52/28). The caregivers reported low BPSD (mean = 24.74/96) with a lower level of caregiver distress (mean = 13.00/60) compared with the severity of dementia symptoms (mean = 11.74/36). On average, caregiver burden was reported to be at the low-to-medium level (mean = 30.43/88) with the level of its components also ranging from low to medium.

The differences in the average scores of caregiver burden based on the independent variables and the results from the bivariate analysis are presented in Table 3. Some factors were found as the correlates of caregiver burden. The gender of the PWDs, the educational level of the caregivers, social support, and BPSD were statistically significantly associated with caregiver burden (*p* < 0.05). Meanwhile, other factors, such as the age and marital status of both the PWDs and caregivers, the gender of the caregivers, the educational level of the PWDs, employment status, family income, the duration of dementia, the duration of caregiving, the caregivers’ relation to the PWDs, caregiving for other family members, and health insurance ownership among the PWDs were not associated with caregiver burden.

Table 4 presents the multivariate analysis of the factors associated with caregiver burden. Two models are displayed, namely (1) the initial model which included all the independent variables simultaneously in the analysis, and (2) the final model which only consisted of the independent variables that were statistically significantly associated with caregiver burden. Both models met the assumptions of linear regression, including the residuals that were normally distributed (*p* > 0.05 for the Shapiro–Wilk W test), no heteroscedasticity (*p* > 0.05 for the Breusch–Pagan test), and no multicollinearity (variance inflation factor (VIF) < 10). The adjusted R-squared value for the final model was slightly lower than the initial model (27.78% vs. 28.99%). However, the final model was the better one, which fits with the data, indicated by the smaller values of the Bayesian information criterion (BIC).

Based on the final multivariate model, there were four factors associated with caregiver burden, namely the gender of the PWDs, the educational level of the caregivers, social support, and BPSD. The caregivers of female PWDs reported more burden than the caregivers of male PWDs (β = 5.58; 95%CI = 2.16; 8.99). In addition, the caregivers who completed high school education and above were observed with less burden than their counterparts who completed less than high school education (β = −6.41; 95%CI = −10.07; −2.74). The increased perceived social support was negatively associated with caregiver burden. For every one score increase in social support, the caregiver’s burden would decrease by 0.26 (β = −0.26; 95%CI = −0.42; −0.10). Meanwhile, BPSD was associated with caregiver burden in a positive direction, of which, caregiver burden would elevate by 0.34 for every additional one-unit score of BPSD (β = 0.34; 95%CI = 0.25; 0.43). Based on the adjusted R-squared value, these four factors in the final model contributed 27.78% to the variance of caregiver burden.

## 4. Discussion

To the best of our knowledge, this study is among the first to explore caregiver burden among older PWDs in Indonesia and the factors associated with the burden. Overall, our findings demonstrated a low-to-medium level of caregiver burden in almost all aspects (i.e., relationship, emotional well-being, social and family life, control of life, and finances). Consistent with these findings, previous studies investigating the burden among caregivers in Taiwan [12] and caregivers from multiple Asian countries [23] reported similar findings of low-to-medium caregiver burden. The social context might explain these results since caregiver burden is related to subjective appraisals. In the Indonesian society, caregivers who have been raised with collectivist values and strong connections between family members might perceive caregiving as a natural and prepared part of their life. Family caregivers may adopt some positive attitudes toward the caregiving experience. 

The burden among caregivers seems an inevitable situation during the caregiving trajectory, particularly when caring for older PWDs; and a challenging situation may arise in various aspects. As supported by the prior literature, caring for patients with dementia might have negative consequences for caregivers such as emotional problems, decreasing social life, work impairment, and increased healthcare service utilisation [24,25,26,27]. Therefore, understanding what factors might contribute to the burden in the caregiving of older PWDs is important to design intervention in a targeted manner. The findings from our study suggest that some factors were statistically significantly associated with caregiver burden, such as the gender of PWDs, the educational level of caregivers, social support, and BPSD.

### 4.1. Factors Associated with Caregiver Burden

Out of the several socio-demographic caregiver characteristics, the gender of PWDs appeared as a factor associated with caregiver burden. This study reported that the caregivers of female PWDs had a greater burden than the caregivers of male PWDs. This perhaps can be explained by the gender differences in the behavioural and psychological symptoms, although this finding was not substantiated in other studies. Female PWDs exhibited more behavioural and psychological symptoms. Previous studies found that female PWDs displayed more delusions [28,29,30], disinhibition [28,29,31], euphoria [30], hallucinations [32], depression [30,32], anxiety [32], and apathy [30]. This fact supports the increasing burden experienced by female PWD caregivers. Another explanation may be related to the gender roles in this population since women are typically caregivers rather than care recipients. To some extent, caring for a female PWD might be perceived as a greater burden than caring for a male PWD, irrespective of the caregivers’ gender. However, further investigation is needed to deepen our understanding of the role of PWDs’ gender on caregiver burden. 

The educational level of caregivers was also found to predict caregiver burden. The caregivers with lower educational levels expressed a higher burden. These findings could be explained by the fact that a low level of education has been associated with low health literacy [33,34]. Low health literacy may affect those caregivers who experience barriers to accessing, understanding, and enacting information about dementia caregiving. As a result, inadequate caregiver health literacy has the potential to contribute to a lack of knowledge. Furthermore, the caregivers with lower levels of health literacy may have barriers to communicating with healthcare professionals and poorer access to caregiver support services [33], which in turn, compound the caregiver burden. 

This study did not find the influence of other characteristics of caregivers, such as their age, gender, and relationship with PWDs, on the self-reported burden in caring for older PWDs. This might be explained by the sociocultural context in many Asian countries, including Indonesia, where caring for older people is perceived as voluntary work. A previous qualitative study from Indonesia indicated that family members tend to express obligation and calling (i.e., intrinsic passion) as a reason for taking up a caregiving task [7]. This expression is strongly related to the implementation of religious values as part of their day-to-day life and identity. Therefore, the motivation in providing care and the extent of the perceived caregiver burden might be perhaps more elucidated by the personal and community values and norms rather than the socio-demographic characteristics of caregivers. However, this warrants further investigation. 

For the caregivers in this study, social support contributed substantially to diminishing their burden. This study verified that social support is a protective factor based on the Stress Process Model [35]. The findings from a recent meta-analysis also found the importance of perceived social support in reducing the subjective caregiver burden in caregiving older adults [36]. In this study, the highest support was received from family members and significant others. The presence of other people in the family who can provide support might help caregivers to appraise the caregiving-related situations as less stressful. Caregivers may have time to break from their caregiving role and alternate to take care of PWDs, which in turn, can reduce the burden of caregiving. In addition, the caregivers with more physical social support will have more time and energy to focus on the caregiving task since others can help with other caring responsibilities. Moreover, the perceived social support reported by caregivers might not only depict a solution for addressing caregiving-related problems, but this might also include reducing the perceived importance of these problems and providing a distraction from these problems [36].

This study demonstrated the association between caregiver burden and BPSD. The caregiver burden increased by 0.34 for one additional BPSD score, similar to a previous study indicating that a high BPSD score contributed 0.27 more to the burden score [37]. Dementia symptoms, including the behavioural and psychological aspects, have a significant influence on caregiver distress. Similar studies with the NPI instrument showed a positive correlation with caregiver burden [38,39]. Delusions, agitation, irritability and nocturnal behaviour, hallucinations, depression, apathy, disinhibition, and motor disturbances are common in patients with dementia. It becomes a burden that leads to the unfavourable psychological and physical well-being of caregivers [40].

The results of this study indicated that caregiver burden was predicted by the caregivers’ perception of social support, BPSD, caregiver education level, and the gender of PWDs. Among the four influencing factors, the perception of the caregivers’ social support is most likely to be modified. Family, friends, and the significant others around the caregiver can protect the caregiver from any perceived stress while caring for PWDs. This needs to be realised by caregivers that family members can be a source of moral support, a distraction, and a place to deal with stressful situations. Additionally, caregivers need adequate knowledge about BPSD and its progression. In the early stages of the disease, BPSD may still be mild and manageable. However, along with the severity of dementia, several types of BPSD can coexist, demanding the fulfilment of complex caregiver needs and care. Sufficient knowledge about BPSD can help caregivers prepare themselves physically and mentally.

There are several implications for clinical practices that we can draw from this study. This study identified specific factors that place some caregivers at higher risk for burden. Our findings indicated that health professionals can implement a screening tool that integrates risk factors with protective factors to identify caregiver burden. The use of such a tool would help health professionals to identify vulnerable caregivers and provide appropriate and timely interventions. The results of this study also revealed that social support is a protective factor that shields caregivers from the burden; therefore, this finding underlines the need to design and implement support programs for caregivers. This may increase their social problem-solving skills, which in turn, can sustain confidence in caregivers’ capacity to manage the stressors due to BPSD and avert negative health consequences. In addition, most caregivers in this study provided care for the PWDs at the early stages of dementia. Future studies should be carried out among the family of the caregivers who care for PWDs in the middle and advanced stages of the disease. Such studies will help health professionals understand the caregiving and burden experienced among caregivers in varied stages of dementia progression.

### 4.2. Study Limitations

Our study design is predisposed to several limitations. First, the study was cross-sectional in design, and therefore, a causal relationship cannot be deduced, and the findings might be susceptible to reverse causality. All the data reported in this study relied on self-report, which is prone to bias, and we used validated questionnaires to improve the quality of all the data we collected. In addition, our findings might be subjected to selection bias since we only recruited the caregivers of PWDs from tertiary hospitals. Furthermore, some variables that might be associated with caregiver burden were not investigated in this study, such as the types and severity of dementia, and the independent daily activities of PWDs. Despite these limitations, this is a novel and important study that captures the caregiver burden of older PWDs in the Indonesian context. While our study helps to define this population of caregivers, future studies that investigate the caregivers’ perceptions of the neuropsychiatric symptoms of PWDs, understand the burden among the caregivers of PWDs in the middle and advanced stages of dementia, and identify strategies for care will allow health professionals to provide appropriate care for patients and caregivers alike. 

## 5. Conclusions

This cross-sectional study of the caregivers of older PWDs showed that caregivers experience burden, and that caregiver burden was predicted by the social support the caregiver perceived, the educational level of the caregiver, BPSD, and the gender of PWDs. As this PWD population might continue to grow, it will be important for future research to identify effective strategies and resources for mitigating the caregiver burden. Additionally, the identification of the specific type of BPSD and how the different stages of dementia cause the burden on caregivers need to be investigated so that the treatment of depression in caregivers can be achieved. By doing so, healthcare professionals can adapt their practice to detect and address the caregiver burden among the caregivers of older PWDs.

## Figures and Tables

**Table 1 ijerph-19-12437-t001:** Characteristics of PWDs and caregivers (*n* = 207).

Variables	PWDs *n* (%)	Caregivers *n* (%)
Age of PWDs		
Min; max	60; 92	
Mean; SD	69.91; 7.57	
60–74 years	151 (72.95)	
≥75 years	56 (27.05)	
Age of caregivers		
Min; max	20; 92	
Mean; SD	51.76; 13.84	
<60 years	144 (69.57)	
≥60 years	63 (30.43)	
Gender		
Male	116 (56.04)	52 (25.12)
Female	91 (43.96)	155 (74.88)
Educational level		
<High school	100 (48.31)	64 (30.92)
≥High school	107 (51.69)	143 (69.08)
Marital status		
Single	3 (1.45)	13 (6.28)
Married	148 (71.50)	189 (91.30)
Divorced/widowed/separated	56 (27.05)	5 (2.42)
Employment status		
Unemployed		118 (57.00)
Employed		89 (43.00)
Family monthly income		
(1 USD = 14,400 rupiah)		
Min; max		2,000,000; 27,000,000
Mean; SD		4,301,932; 4,431,791
<4,000,000 rupiah		125 (60.39)
≥4,000,000 rupiah		82 (39.61)
Duration of diagnosis with dementia (in months)		
Min; max	6; 84	
Mean; SD	24.76; 20.22	
<24 months	111 (53.62)	
≥24 months	96 (46.38)	
Duration of caregiving (in months)		
Min; max		6; 84
Mean; SD		23.28; 18.94
<24 months		155 (55.56)
≥24 months		92 (44.44)
Relation to PWDs		
Adult children/daughter-in-law		102 (49.28)
Spouses		94 (45.41)
Relatives		11 (5.31)
Caregiving for other family members		
No		189 (91.30)
Yes		18 (8.70)
Having health insurance for PWDs		
No	14 (6.67)	
Yes	193 (93.24)	

PWDs = patients with dementia; Min = minimum value; Max = maximum value; SD = standard deviation.

**Table 2 ijerph-19-12437-t002:** Perceived social support, BPSD, and caregiver burden (*n* = 207).

Variables	Possible Range Min; Max	Mean (SD)
Social support (*total*)	12; 84	65.48 (10.49)
Family	4; 28	23.52 (3.80)
Friends	4; 28	18.30 (5.64)
Significant others	4; 28	23.66 (4.08)
BPSD (*total*)	0; 96	24.74 (18.10)
NPI-Q severity of dementia symptoms	0; 36	11.74 (7.95)
NPI-Q caregiver distress	0; 60	13.00 (10.49)
Caregiver burden (*total*)	0; 88	30.43 (14.51)
Burden in the relationship	0; 24	10.68 (4.91)
Emotional well-being	0; 28	10.22 (4.59)
Social and family life	0; 16	4.03 (3.34)
Finances	0; 4	1.07 (1.14)
Loss of control over one’s life	0; 16	4.43 (2.97)

BPSD = behavioural and psychological symptoms of dementia; NPI-Q = Neuropsychiatric Inventory Questionnaire; Min = minimum value; Max = maximum value; SD = standard deviation.

**Table 3 ijerph-19-12437-t003:** The differences in average scores of caregiver burden by independent variables and bivariate analysis of factors associated with caregiver burden (*n* = 207).

Variables	Mean; SD	β (95% CI)	*p*-Value
Age of PWDs			
60–74 years	30.54; 14.34	Ref	
≥75 years	30.16; 15.08	−0.38 (−4.86; 4.11)	0.869
Age of caregivers			
<60 years	30.79; 13.95	Ref	
≥60 years	29.62; 15.81	−1.17 (−5.50; 3.16)	0.594
Gender of PWDs			
Male	27.75; 14.66	Ref	
Female	33.86; 13.63	6.11 (2.18; 10.03)	0.002
Gender of caregivers			
Male	31.10; 14.58	Ref	
Female	30.21; 14.53	−0.88 (−5.48; 3.71)	0.705
Educational level of PWDs			
<High school	32.09; 14.87	Ref	
≥High school	28.89; 14.05	−3.20 (−7.17; 0.76)	0.113
Educational level of caregivers			
<High school	34.23; 15.77	Ref	
≥High school	28.73; 13.62	−5.50 (−9.75; −1.25)	0.011
Marital status of PWDs			
Single	41.67; 11.15	Ref	
Married	28.08; 14.80	−13.59 (−29.77; 2.59)	0.099
Divorced/widowed/separated	36.05; 12.02	−5.61 (−22.05; 10.83)	0.502
Marital status of caregivers			
Single	27.69; 13.55	Ref	
Married	30.81; 14.72	3.12 (−5.09; 11.32)	0.455
Divorced/widowed/separated	23.40; 4.34	−4.29 (−19.35; 10.77)	0.575
Employment status			
Unemployed	28.95; 13.81	Ref	
Employed	32.41; 15.24	3.46 (−0.54; 7.45)	0.090
Family monthly income			
<4,000,000	31.43; 14.70	Ref	
≥4,000,000	28.91; 14.17	−2.52 (−6.58; 1.54)	0.223
Duration of diagnosis with dementia			
<24 months	30.55; 14.29	Ref	
≥24 months	30.30; 14.83	−0.25 (−4.24; 3.75)	0.903
Duration of caregiving			
<24 months	30.99; 14.43	Ref	
≥24 months	29.74; 14.65	−1.25 (−5.26; 2.75)	0.539
Relation to PWDs			
Adult child/daughter in law	31.17; 13.77	Ref	
Partners	29.70; 15.63	−1.46 (−5.57; 2.64)	0.483
Relatives	29.91; 11.68	−1.26 (−10.37; 7.85)	0.786
Caregiving for other family members			
No	30.70; 14.49	Ref	
Yes	27.67; 14.80	−3.03 (−10.09; 4.03)	0.398
Having health insurance for PWDs			
No	28.57; 13.43	Ref	
Yes	30.57; 14.61	2.00 (−5.93; 9.93)	0.620
Social support		−0.27 (−0.46; −0.09)	0.004
BPSD		0.34 (0.24; 0.44)	<0.001

PWDs = patients with dementia; BPSD = behavioural and psychological symptoms of dementia; SD = standard deviation; β = regression coefficient; CI = confidence interval.

**Table 4 ijerph-19-12437-t004:** Multivariate analysis of factors associated with caregiver burden (*n* = 207).

Variables	Initial Model		Final Model	
	aβ (95% CI)	*p*-Value	aβ (95% CI)	*p*-Value
Age of PWDs *(Ref: 60–74 years)*				
≥75 years	−1.02 (−5.33; 3.30)	0.643		
Age of caregivers *(Ref: < 60 years)*				
≥60 years	1.21 (−4.29; 6.71)	0.665		
Gender of PWDs *(Ref: Male)*				
Female	4.01 (−0.79; 8.80)	0.101	5.58 (2.16; 8.99)	0.001
Gender of caregivers *(Ref: Male)*				
Female	2.28 (−2.45; 7.02)	0.343		
Educational level of PWDs *(Ref: < High school)*				
≥High school	−0.12 (−4.49; 4.26)	0.956		
Educational level of caregivers *(Ref: < High school)*				
≥High school	−4.27 (−9.07; 0.52)	0.080	−6.41 (−10.07; −2.74)	0.001
Marital status of PWDs *(Ref: Single)*				
Married	−11.63 (−28.66; 5.40)	0.180		
Divorced/widowed/separated	−5.30 (−22.20; 11.60)	0.537		
Marital status of caregiver *(Ref: Single)*				
Married	4.83 (−2.91; 12.57)	0.219		
Divorced/widowed/separated	−0.12 (−13.38; 13.14)	0.986		
Employment status *(Ref: Unemployed)*				
Employed	4.36 (0.38; 8.34)	0.032		
Family monthly income *(Ref: < 4,000,000)*				
≥4,000,000	−3.43 (−7.68; 0.83)	0.114		
Duration of diagnosis with dementia *(Ref: < 24 months)*				
≥24 months	9.32 (−4.37; 23.00)	0.181		
Duration of caregiving *(Ref: < 24 months)*				
≥24 months	−9.34 (−22.95; 4.27)	0.177		
Relation to PWDs *(Ref: Adult child/daughter-in-law)*				
Partners	3.01 (−3.42; 9.44)	0.357		
Relatives	−5.84 (−14.46; 2.79)	0.183		
Caregiving for other family member *(Ref: No)*				
Yes	−2.04 (−8.57; 4.48)	0.538		
Having health insurance for PWDs *(Ref: No)*				
Yes	1.16 (−6.01; 8.34)	0.749		
Social support	−0.22 (−0.39; −0.06)	0.009	−0.26 (−0.42; −0.10)	0.002
BPSD	0.33 (0.23; 0.44)	<0.001	0.34 (0.25; 0.43)	<0.001
**Adjusted R-squared**	**28.99%**		**27.78%**	
**Bayesian information criterion (BIC)**	**1713.75**		**1648.99**	
**Shapiro–Wilk W test for normality of residuals (*p*-value)**	**0.167**		**0.07**	
**Breusch-Pagan test for heteroscedasticity (*p*-value)**	**0.336**		**0.470**	
**Mean of variance inflation factor (VIF)**	**5.01**		**1.00**	

PWDs = patients with dementia; BPSD = behavioural and psychological symptoms of dementia; aβ = adjusted regression coefficient; CI = confidence interval.

## Data Availability

The datasets for this study are available from the corresponding author upon a reasonable request.

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
