# Peer review of "Factors Associated with Caregiver Burden in Caregivers of Older Patients with Dementia in Indonesia"

_ijerph, 2022, doi:10.3390/ijerph191912437_

Round 1
Reviewer 1 Report
In the manuscript, "Factors associated with caregiver burden in caregivers of older patients with dementia in Indonesia", the authors address an important issue. Specifically, given that we have an aging population that will involve a greater proportion of the population with dementia, we will have a greater number of informal caregivers caring for persons with dementia. Understanding the costs of this type of care is imperative.
While the authors address an important topic, there are some edits that might improve the manuscript.
Introduction
Page 1, Line 39 The authors may want to select a term that is not perceived as having negative connotations as "elderly". Terms such as older adults or aging adults.
Page 2, Line 57 You can remove the word "since" and the comma that follows
Page 2, Line 71 You can remove the word "also"
Methods
Page 2, Line 97 The authors write, "Caregivers who had cognitive impairment with a total score > 7 from a Six-item Cognitive Impairment Test (6CIT) were excluded from this study." If one of the goals of this study is to shed light on caregiver issues, it would be helpful to know how many caregivers were excluded due to cognitive impairment.
Results
Page 4, Line 161 The sentence, "The majority of PWDs dan caregivers completed at least..." includes the word "dan". I am not sure why it is there.
Discussion
Page 9, Line 238 The authors sometimes hyphenate sociodemographic and sometimes do not. Please write it out consistently throughout the manuscript
Page 9 Lines 238-245 To explain the differences in gender in burden, the authors write, "Out of the several sociodemographic caregiver characteristics, gender of PWDs appeared as a factor associated with caregiver burden. This study reported that caregivers 239 of female PWDs had a greater burden than caregivers of male PWDs. This perhaps can be explained by gender differences in behavioural and psychological symptoms, although this finding was not substantiated in other studies. This fact supports the increasing suffering experienced by female PWD caregivers." I would like to authors to clarify for me whether there are alternative explanations, such as traditional gender roles. For example, are there expected gender roles in this population that women are typically caregivers rather than care recipients and therefore caring for a female is perceived as a greater burden than caring for a male (independent of whether the caregiver is a female daughter, male spouse, female spouse, etc.)? I would also be cautious about using the word "suffering". It is not implicit that because something is a burden that it is equivalent to "suffering". We provide care for children in society and do not usually describe that as "suffering". If the term is only used to describe caring for older adults, it may be perceived as ageist.
Author Response
Reviewer 1
In the manuscript, "Factors associated with caregiver burden in caregivers of older patients with dementia in Indonesia", the authors address an important issue. Specifically, given that we have an aging population that will involve a greater proportion of the population with dementia, we will have a greater number of informal caregivers caring for persons with dementia. Understanding the costs of this type of care is imperative.
Response:
We would like to thank you for your feedback on this manuscript. We have revised the manuscript according to your comments.
While the authors address an important topic, there are some edits that might improve the manuscript.
Introduction
Page 1, Line 39 The authors may want to select a term that is not perceived as having negative connotations as "elderly". Terms such as older adults or aging adults.
Response:
We have changed elderly to older adults.
Page 2, Line 57 You can remove the word "since" and the comma that follows
Response:
We have removed the word “since”.
Page 2, Line 71 You can remove the word "also"
Response:
We have removed the word “also”.
Methods
Page 2, Line 97 The authors write, "Caregivers who had cognitive impairment with a total score > 7 from a Six-item Cognitive Impairment Test (6CIT) were excluded from this study." If one of the goals of this study is to shed light on caregiver issues, it would be helpful to know how many caregivers were excluded due to cognitive impairment.
Response:
We have added this information to the manuscript.
“Ten participants who had cognitive impairment with a total score > 7 from a Six-item Cognitive Impairment Test (6CIT) were excluded from this study.”
Results
Page 4, Line 161 The sentence, "The majority of PWDs dan caregivers completed at least..." includes the word "dan". I am not sure why it is there.
Response:
We have revised this sentence.
Discussion
Page 9, Line 238 The authors sometimes hyphenate sociodemographic and sometimes do not. Please write it out consistently throughout the manuscript
Response:
We used “socio-demographic” throughout the manuscript.
Page 9 Lines 238-245 To explain the differences in gender in burden, the authors write, "Out of the several sociodemographic caregiver characteristics, gender of PWDs appeared as a factor associated with caregiver burden. This study reported that caregivers 239 of female PWDs had a greater burden than caregivers of male PWDs. This perhaps can be explained by gender differences in behavioural and psychological symptoms, although this finding was not substantiated in other studies. This fact supports the increasing suffering experienced by female PWD caregivers." I would like to authors to clarify for me whether there are alternative explanations, such as traditional gender roles. For example, are there expected gender roles in this population that women are typically caregivers rather than care recipients and therefore caring for a female is perceived as a greater burden than caring for a male (independent of whether the caregiver is a female daughter, male spouse, female spouse, etc.)? I would also be cautious about using the word "suffering". It is not implicit that because something is a burden that it is equivalent to "suffering". We provide care for children in society and do not usually describe that as "suffering". If the term is only used to describe caring for older adults, it may be perceived as ageist.
Response:
We would like to thank you for your positive comments. We have added explanations based on traditional gender roles. We also changed the term “suffering” to “burden”.
“Another explanation may be related to gender roles in this population since women are typically caregivers rather than care recipients. To some extent, caring for a female PWD might be perceived as a greater burden than caring for a male PWD, irrespective of caregivers’ gender. However, further investigation is needed to deeper our understanding of the role of PWDs’ gender on caregiving burden.”
Reviewer 2 Report
Many thanks for providing me with the opportunity of reviewing the manuscript. This paper mainly examines the correlates of care burden among caregivers of people with dementia. In general, the topic of this paper is significant in the field. The structure is well written. My concerns are as follow:
Introduction
1. When socio-demographic and caregiver-related factors were considered, the authors may need to address more contextual issues which affect care burden in Indonesia. For example, how the impacts of religion, culture, or health insurance system work on care burden? Do genders matter in the field? Any differences in Java compared to other regions in Indonesia (p.2)?
Methods
2. Did the participants fill the questionnaires by themselves? Or someone such as research assistants helped them? If they were unable to read, write, listen, or had language barriers, would they be excluded? Who conducted 6CIT for caregivers? (p.3)
3. If possible, please provide evidences of psychometrics for these scales (ZBI, MSPSS, NPI-Q) in the population of Indonesia. What is the language used in the study? Did you have local language versions of the scales?
Results
4. The authors should try to avoid repetition in the paragraph and the table about the descriptive findings. (p.4)
5. Regarding the relationships, Table 1 showed 3 categories: adult children/daughter-in-law, spouses, and relatives (should be other relatives). How about son-in-law? Was it classified in to others? Why? (p.5)
6. Table 2 looks confusing. You may need to re-arrange the format to make it clear.
7. In my opinion, Table 3 is not necessary because Table 4 has showed the same significant differences such as gender of PWDs, educational level of caregivers, social support, and BPSD.
Discussion
8. You may consider to compare caregiver burden with other countries when low-to-medium level of caregiver burden was reported (p.9 line 226).
9. Regarding the gender of PWDs, the explanation in the article is not convincing to me. Even the behavioural and psychological symptoms were controlled in the regression model, there was still significant association between gender of PWDs and caregiver burden (p.9 line 240-246).
10. When the related factors were identified, the clinical and research implication of the study findings should be addressed.
11. Regarding the study limitations, several points should be considered. Recall bias was mentioned. How about selection bias? Participants who visited tertiary hospitals could be representatives of all caregivers in Indonesia? What about causal relationships? For example, heavy care burden makes caregivers get less social support? Did you have other important factors which were not included in the study? Severity of dementia (ex. clinical dementia rating), dementia types, independent activities of daily living? Will they confound the study results?
Author Response
Reviewer 2
Many thanks for providing me with the opportunity of reviewing the manuscript. This paper mainly examines the correlates of care burden among caregivers of people with dementia. In general, the topic of this paper is significant in the field. The structure is well written. My concerns are as follow:
Response:
Thank you for your feedback on this manuscript. We have revised the manuscript according to your comments.
Introduction
- When socio-demographic and caregiver-related factors were considered, the authors may need to address more contextual issues which affect care burden in Indonesia. For example, how the impacts of religion, culture, or health insurance system work on care burden? Do genders matter in the field? Any differences in Java compared to other regions in Indonesia (p.2)?
Response:
Thank you for your input. We have added more information regarding the context of our study.
“The caregiving role has been traditionally viewed as a woman’s responsibility in Indonesia, similar to many countries, especially when the care recipients are parents. In addition, findings from an ethnography study in East Java and West Sumatra demonstrated the preference of a daughter to fulfil the caring responsibility of older adults [6]. Caring for the infirm members of the household is considered a cultural duty for the whole family members in the Indonesian society which is also in line with religious values. Being a caregiver of their loved one might be interpreted as gratitude among caregivers. This also reaffirms the emotional bond between them, as well as perceived as a natural consequence of kinship ties and moral values [7]. It is a way to show devotion and affection to the care recipients. Strong kinship ties can be diminished the level of caregiver stress.”
Methods
- Did the participants fill the questionnaires by themselves? Or someone such as research assistants helped them? If they were unable to read, write, listen, or had language barriers, would they be excluded? Who conducted 6CIT for caregivers? (p.3)
Response:
We have added this information to the manuscript.
“If potential participants or caregivers were over 60 years old, the researcher or research assistant screened them for cognitive function. Ten participants who had cognitive impairment with a total score > 7 from a Six-item Cognitive Impairment Test (6CIT) were excluded from this study. Two hundred and seven caregivers who provided care for older PWDs aged 60 years or more were included in this present study. Participants provided written informed consent prior to participating in the study. If participants had difficulties in completing the questionnaires, the research or research assistants would help them to have all questions completed.”
- If possible, please provide evidences of psychometrics for these scales (ZBI, MSPSS, NPI-Q) in the population of Indonesia. What is the language used in the study? Did you have local language versions of the scales?
Response:
We have added the psychometric properties of scales that we used in this study from a previous study conducted in Indonesia. We also added information that we translated these scales from English into the Indonesian language (Bahasa).
Results
- The authors should try to avoid repetition in the paragraph and the table about the descriptive findings. (p.4)
Response:
We have revised the manuscript accordingly.
- Regarding the relationships, Table 1 showed 3 categories: adult children/daughter-in-law, spouses, and relatives (should be other relatives). How about son-in-law? Was it classified in to others? Why? (p.5)
Response:
We have added information in this regard.
“We found no son-in-law providing care for PWDs in our study.”
- Table 2 looks confusing. You may need to re-arrange the format to make it clear.
Response:
We have revised Table 2.
- In my opinion, Table 3 is not necessary because Table 4 has showed the same significant differences such as gender of PWDs, educational level of caregivers, social support, and BPSD.
Response:
Even though both Tables 3 and 4 show the same findings in terms of what variables were statistically significantly associated with caregiving burden, both tables have different purposes. Table 3 presents the unadjusted associations (bivariate analyses), whilst Table 4 shows findings from the adjusted association (multivariate analyses). In addition, in Table 3, we also presented descriptive findings in terms of difference scores of caregiving burden by independent variables that would be helpful for readers in understanding our paper. Therefore, we decided to keep Table 3 in our manuscript.
Discussion
- You may consider to compare caregiver burden with other countries when low-to-medium level of caregiver burden was reported (p.9 line 226).
Response:
We have added findings from previous settings that found similar level of caregiver burden.
“Consistent with these findings, previous studies investigating burden among caregivers in Taiwan [12] and caregivers from multiple Asian countries [23] reported similar findings on low-to-medium caregiver burden.”
- Regarding the gender of PWDs, the explanation in the article is not convincing to me. Even the behavioural and psychological symptoms were controlled in the regression model, there was still significant association between gender of PWDs and caregiver burden (p.9 line 240-246).
Response:
We have added information based on gender roles within the context of Indonesia that might help explain our findings.
“Another explanation may be related to gender roles in this population since women are typically caregivers rather than care recipients. To some extent, caring for a female PWD might be perceived as a greater burden than caring for a male PWD, irrespective of caregivers’ gender. However, further investigation is needed to deeper our understanding of the role of PWDs’ gender on caregiving burden.”
- When the related factors were identified, the clinical and research implication of the study findings should be addressed.
Response:
We have added the implications of our findings.
“The results of this study indicated that caregiver burden was predicted by caregivers’ perception of social support, BPSD, caregiver education level, and gender of PWDs. Among the four influencing factors, the perception of caregivers' social support is most likely to be modified. Family, friends, and the significant others around the caregiver can be a protector from the stress perceived by the caregiver while caring for PWDs. This needs to be realised by caregivers that family members can be a source of moral support, a distraction, and a place to deal with stressful situations. Also, caregivers need adequate knowledge about BPSD and its progression. In the early stages of the disease, BPSD may still be mild and manageable. However, along with the severity of dementia, several types of BPSD can coexist, demanding the fulfilment of complex caregiver needs and care. Sufficient knowledge about BPSD can help caregivers prepare themselves physically and mentally.
There are several implications for clinical practices that we can draw from this study. This study identified specific factors that place some caregivers at higher risk for burden. Findings indicated that health professionals can implement a screening tool that integrates risk factors with protective factors to identify caregiver burden. The use of such a tool would help health professionals to identify vulnerable caregivers and provide appropriate and timely interventions. The results of this study also revealed that social support is a protective factor that shields caregivers from the burden, therefore this finding underlines the need to design and implement support programs for caregivers. This may increase social-problem solving skills, which in turn, can sustain confidence in caregivers’ capacity to manage stressors due to BPSD and avert negative health consequences. In addition, most caregivers in this study provided care for PWDs at the early stages of dementia. Future studies should carry out among family caregivers who care for PWDs in the middle and advanced stages of the disease. Such studies will help health professionals understand the caregiving and burden experienced among caregivers in varied stages of dementia progression.”
- Regarding the study limitations, several points should be considered. Recall bias was mentioned. How about selection bias? Participants who visited tertiary hospitals could be representatives of all caregivers in Indonesia? What about causal relationships? For example, heavy care burden makes caregivers get less social support? Did you have other important factors which were not included in the study? Severity of dementia (ex. clinical dementia rating), dementia types, independent activities of daily living? Will they confound the study results?
Response:
Thank you for your positive comments. We have added some limitations of our study based on your comments.
“Our study design is predisposed to several limitations. First, the study was cross-sectional in design, and therefore, the causal relationship cannot be deduced, and findings might be susceptible to reverse causality. All the data reported in this study relied on self-report, which is prone to bias, and we used validated questionnaires to improve the quality of all the data we collected. In addition, findings might be subjected to selection bias since we only recruited caregivers of PWDs from tertiary hospitals. Furthermore, some variables that might be associated with caregiver burden were not investigated in this study, such as types and severity of dementia, and independent daily activities of PWDs. Despite these limitations, this is a novel and important study that captures the caregiver burden of older PWDs in the Indonesian context. While our study helps to define this population of caregivers, future studies that investigate caregivers’ perceptions of neuropsychiatric symptoms of PWDs, understand the burden among caregivers of PWDs in the middle and advanced stages of dementia, and identify strategies for care will allow health professionals to provide appropriate care for patients and caregivers alike.”
Reviewer 3 Report
The text is correctly edited in terms of content and form. Research tools and literature have been selected optimally.
The authors could consider extending the section on ethical aspects of the research. It was limited to citing the legal acts and codes of ethics. The ethical principles could be described in more detail (e.g. content of informed consent).
Author Response
Reviewer 3
The text is correctly edited in terms of content and form. Research tools and literature have been selected optimally. The authors could consider extending the section on ethical aspects of the research. It was limited to citing the legal acts and codes of ethics. The ethical principles could be described in more detail (e.g. content of informed consent).
Response:
We thank you for your feedback. We have added more information on the Ethics approval and consent to participate.
“Before participating in this study, participants (caregivers) were provided with information about the background and aim of the study, the benefits and risks, and compensation they would receive for their participation. In addition, participants were also informed about their rights to withdraw from this study at any time without any consequences. Written informed consent was given by all participating caregivers.”
Reviewer 4 Report
In order to make the paper more effective in proposing possible interventions, the authors should address the question of what kind of services may help caregivers. I mean that the four influencing factors should be discussed in light of what in practice can be done to support caregivers (taking into account the welfare system of the country).
Author Response
Reviewer 4
In order to make the paper more effective in proposing possible interventions, the authors should address the question of what kind of services may help caregivers. I mean that the four influencing factors should be discussed in light of what in practice can be done to support caregivers (taking into account the welfare system of the country).
Response:
We thank you for your feedback. We have added more information on the implications of the findings.
“The results of this study indicated that caregiver burden was predicted by caregivers’ perception of social support, BPSD, caregiver education level, and gender of PWDs. Among the four influencing factors, the perception of caregivers' social support is most likely to be modified. Family, friends, and the significant others around the caregiver can be a protector from the stress perceived by the caregiver while caring for PWDs. This needs to be realised by caregivers that family members can be a source of moral support, a distraction, and a place to deal with stressful situations. Also, caregivers need adequate knowledge about BPSD and its progression. In the early stages of the disease, BPSD may still be mild and manageable. However, along with the severity of dementia, several types of BPSD can coexist, demanding the fulfilment of complex caregiver needs and care. Sufficient knowledge about BPSD can help caregivers prepare themselves physically and mentally.
There are several implications for clinical practices that we can draw from this study. This study identified specific factors that place some caregivers at higher risk for burden. Findings indicated that health professionals can implement a screening tool that integrates risk factors with protective factors to identify caregiver burden. The use of such a tool would help health professionals to identify vulnerable caregivers and provide appropriate and timely interventions. The results of this study also revealed that social support is a protective factor that shields caregivers from the burden, therefore this finding underlines the need to design and implement support programs for caregivers. This may increase social-problem solving skills, which in turn, can sustain confidence in caregivers’ capacity to manage stressors due to BPSD and avert negative health consequences. In addition, most caregivers in this study provided care for PWDs at the early stages of dementia. Future studies should carry out among family caregivers who care for PWDs in the middle and advanced stages of the disease. Such studies will help health professionals understand the caregiving and burden experienced among caregivers in varied stages of dementia progression.”